# Integration of pathologic characteristics, genetic risk and lifestyle exposure for colorectal cancer survival assessment

Junyi Xin[1,2,3,11], Dongying Gu[4,11], Shuwei Li[2,3,11], Sangni Qian[5,6,11], Yifei Cheng[2,3], Wei Shao[2,3], Shuai Ben[2,3], Silu Chen[2,3], Linjun Zhu[7], Mingjuan Jin[5,6], Kun Chen[5,6], Zhibin Hu [8], Zhengdong Zhang[2,3], Mulong Du [9,12], Hongbing Shen [8,12] & Meilin Wang [2,3,10,12] ✉

The development of an effective survival prediction tool is key for reducing colorectal cancer mortality. Here, we apply a three-stage study to devise a polygenic prognostic score (PPS) for stratifying colorectal cancer overall survival. Leveraging two cohorts of 3703 patients, we first perform a genome-wide survival association analysis to develop eight candidate PPSs. Further using an independent cohort with 470 patients, we identify the 287 variants-derived PPS (*i.e.*, PPS$_{287}$) achieving an optimal prediction performance [hazard ratio (HR) per SD = 1.99, $P = 1.76 \times 10^{-8}$], accompanied by additional tests in two external cohorts, with HRs per SD of 1.90 ($P = 3.21 \times 10^{-14}$; 543 patients) and 1.80 ($P = 1.11 \times 10^{-9}$; 713 patients). Notably, the detrimental impact of pathologic characteristics and genetic risk could be attenuated by a healthy lifestyle, yielding a 7.62% improvement in the 5-year overall survival rate. Therefore, our findings demonstrate the integrated contribution of pathologic characteristics, germline variants, and lifestyle exposure to the prognosis of colorectal cancer patients.

Colorectal cancer is the third most commonly diagnosed cancer and the second leading cause of cancer death worldwide, with over 1.8 million new cases and 0.9 million deaths in 2020[1]. Remarkably, colorectal cancer is also the most common cause of cancer death in six countries and ranks among the top three leading causes of cancer death in 104 countries[2]. Therefore, there is an urgent clinical need to provide more effective survival prediction tools to reduce colorectal cancer mortality and improve patients' outcome. It is well known that clinical and pathologic characteristics (e.g., clinical stage) are important prognostic factors in predicting survival outcomes[3,4]. In addition, recent studies have suggested that genetic biomarkers also play vital roles in determining the risk of cancer outcomes[5]; for example, one

[1]Department of Bioinformatics, School of Biomedical Engineering and Informatics, Nanjing Medical University, Nanjing, China. [2]Department of Environmental Genomics, Jiangsu Key Laboratory of Cancer Biomarkers, Prevention and Treatment, Collaborative Innovation Center for Cancer Personalized Medicine, School of Public Health, Nanjing Medical University, Nanjing, China. [3]Department of Genetic Toxicology, The Key Laboratory of Modern Toxicology of Ministry of Education, Center for Global Health, School of Public Health, Nanjing Medical University, Nanjing, China. [4]Department of Oncology, Nanjing First Hospital, Nanjing Medical University, Nanjing, China. [5]Department of Epidemiology and Biostatistics at School of Public Health, Zhejiang University School of Medicine, Hangzhou, China. [6]Cancer Institute, The Second Affiliated Hospital, Zhejiang University School of Medicine, Hangzhou, China. [7]Department of Oncology, The First Affiliated Hospital of Nanjing Medical University, Nanjing, China. [8]Department of Epidemiology, Center for Global Health, School of Public Health, Nanjing Medical University, Nanjing, China. [9]Department of Biostatistics, Center for Global Health, School of Public Health, Nanjing Medical University, Nanjing, China. [10]The Affiliated Suzhou Hospital of, Suzhou Municipal Hospital, Gusu School, Nanjing Medical University, Suzhou, China. [11]These authors contributed equally: Junyi Xin, Dongying Gu, Shuwei Li, Sangni Qian. [12]These authors jointly supervised this work: Mulong Du, Hongbing Shen, Meilin Wang. ✉ e-mail: mwang@njmu.edu.cn

study demonstrated the clinical ability of genetic variants for predicting the recurrence and death of renal cell carcinoma[6].

To date, genome-wide association studies (GWASs) have identified over 200 single nucleotide polymorphisms (SNPs) associated with the risk of colorectal cancer[7,8]. Interestingly, these risk-associated variants have contributed to the development of polygenic risk score (PRS), a valuable method that aggregates the modest effect of each SNP, which has been demonstrated to be effective in identifying high-risk individuals of developing colorectal cancer[9–11]. However, the genetic architecture of colorectal cancer survival outcome has not been widely estimated. Noteworthily, survival probability is another critical indicator, that can reflect the tumor burden and prognosis of disease patients[12]. In particular, our previous study demonstrated the limited clinical utility of risk-based PRS in predicting cancer survival, emphasizing that a polygenic prognostic score (PPS) is needed instead for determining the genetic risk of death among colorectal cancer patients[13].

Notably, recent prospective studies have indicated that a healthy lifestyle (e.g., healthy diet) could significantly influence the risk of death among patients with colorectal cancer[14,15]. For example, Zutphen et al. found that improving individual lifestyle after colorectal cancer diagnosis could reduce the risk of all-cause mortality by approximately 20%[15]. However, whether there is a joint effect of pathologic characteristics, genetic risk, and healthy lifestyle on colorectal cancer progression remains unclear.

In this study, we performed a genome-wide survival association meta-analysis of colorectal cancer in East Asian (EAS) and European (EUR) populations; and developed a robust PPS that can be used to stratify colorectal cancer survival; and further evaluated the benefit of adherence to a healthy lifestyle in reducing the risk of death, particularly in the subset of patients with a high pathologic stage or grade, and a high genetic risk.

## Results

### Study design
Here, a three-stage study design was applied (Fig. 1). In the first derivation stage, leveraging two independent colorectal cancer survival GWAS datasets (i.e., NJCRC and UK Biobank cohorts), we performed a meta-analysis to identify survival-associated genetic loci, as well as eight candidate PPSs with different approaches. In the second validation stage, we assessed the discriminatory accuracy of each PPS in an independent longitudinal cohort from The Cancer Genome Atlas (TCGA) to determine an optimal PPS framework for 5-year overall survival prediction. In the third testing stage, using the external ZJCRC cohort and Prostate, Lung, Colorectal and Ovarian (PLCO) cancer screening trial, we further estimated the efficacy of the optimal PPS in colorectal cancer survival prediction, and evaluated the joint effect of pathologic stage or grade, genetic risk and healthy lifestyle (Supplementary Table 1) on the prognosis of colorectal cancer patients.

### Meta-analysis of colorectal cancer survival GWASs
In the derivation stage, leveraging the genetic and clinical data of colorectal cancer patients from NJCRC (1082 cases of EAS ancestry) and UK Biobank (2621 cases of EUR ancestry; Supplementary Fig. 1) cohorts (Table 1), we performed a meta-analysis to identify genetic variants associated with colorectal cancer overall survival (Supplementary Fig. 2A). No residual population stratification was observed (lambda = 1.027; Supplementary Fig. 2B).

Notably, we found two independent variants that were significantly associated with colorectal cancer overall survival beyond the suggestive genome-wide significance ($P_{Cox} < 5 \times 10^{-6}$), namely the rs10967103 [9p21.2; hazard ratio $(HR)_{meta} = 1.70$, $P_{meta} = 4.05 \times 10^{-6}$] and rs79067806 (12q12; $HR_{meta} = 1.89$, $P_{meta} = 4.14 \times 10^{-6}$; Supplementary Table 2; Supplementary Fig. 2C, D). However, there were no SNP-gene expression associations reported in the Genotype-Tissue

Expression (GTEx) project for rs10967103 and rs79067806. In addition, although these two SNPs were located nearby previously reported risk-related regions, they were not observed to be associated with the risk of colorectal cancer in a previous GWAS meta-analysis of case-control studies[9] [35,145 cases and 288,934 controls; rs10967103: odds ratio $(OR)_{meta} = 1.02$, $P_{meta} = 0.449$; rs79067806: $OR_{meta} = 1.00$, $P_{meta} = 0.955$; Supplementary Table 3].

### Construction and validation of PPSs with multiple approaches
Subsequently, we aimed to construct and validate a solid PPS for colorectal cancer survival prediction. Among the eight candidate PPSs (Table 2), seven were significantly associated with an increased risk of all-cause death in the TCGA cohort (470 patients) of EUR ancestry, with HR per standard deviation (SD) increase ranging from 1.47 (P = 0.001) for the clumping and P value thresholding (i.e., C + T) method (parameter of P value: $1 \times 10^{-4}$) to 1.99 ($P = 1.76 \times 10^{-8}$) for the random survival forest (RSF) method.

Notably, the RSF approach-based PPS that harbored 287 SNPs (defined as $PPS_{287}$; Supplementary Data 1) achieved the optimal discriminatory ability for 5-year overall survival prediction, with a time-dependent area under the receiver operating characteristics (ROC) curve (AUC) of 0.652. We then divided the patients into high- and low-PPS groups, with the median score of $PPS_{287}$ as a cut-off value. Compared to patients in the low-PPS group, those carried with high-PPS had shorter overall survival (log-rank P < 0.001) in the validation (i.e., TCGA cohort; Supplementary Fig. 3A) datasets. In addition, the calibration and time-dependent ROC curves of the $PPS_{287}$ model showed good agreement between the predicted and observed 5-year survival probability (Supplementary Fig. 3B), as well as excellent performance in 5-year survival prediction (Supplementary Fig. 3C).

### Testing the optimal PPS in external cohorts
We further evaluated the performance of $PPS_{287}$, the optimal PPS, in two external cohorts, namely the ZJCRC cohort (543 patients of EAS ancestry) and PLCO cohort (713 patients of EUR ancestry). As expected, $PPS_{287}$ was significantly associated with an increased risk of all-cause death in both the ZJCRC (HR per SD = 1.90, $P = 3.21 \times 10^{-14}$) and PLCO (HR per SD = 1.80, $P = 1.11 \times 10^{-9}$; Supplementary Table 4) cohorts. Similar associations were also found between $PPS_{287}$ and 3-year or 5-year colorectal cancer overall survival. The AUCs at 5-year were 0.649 in the ZJCRC cohort and 0.658 in the PLCO cohort, which were similar with the predictive accuracy in the validation cohort (i.e., TCGA).

In addition, using the median score as a cut-off to divide the low- and high-PPS subgroups, patients in the high-PPS group had poorer overall survival than patients carried with low-PPS in the two cohorts (ZJCRC: log-rank $P = 7.68 \times 10^{-9}$; PLCO: log-rank $P = 3.82 \times 10^{-5}$; Fig. 2A). Interestingly, when stratified by clinical factors (e.g., sex, age, smoking status and drinking status), the high-PPS was still broadly and significantly associated with poorer prognosis in the two cohorts (HR > 1; Supplementary Fig. 4A, B). Similar results were also observed in the sensitivity analyses (Supplementary Table 5).

### Additional benefits of PPS to the clinical prognostic model
In the ZJCRC and PLCO cohorts, several clinical factors associated with the overall survival of colorectal cancer were identified (Supplementary Tables 6 and 7), including age (ZJCRC: HR = 1.05, $P = 8.33 \times 10^{-10}$; PLCO: HR = 1.05, $P = 5.21 \times 10^{-5}$), stage (PLCO: $HR_{trend} = 2.82$, $P_{trend} = 4.69 \times 10^{-34}$) and grade (PLCO: $HR_{trend} = 2.53$, $P_{trend} = 2.48 \times 10^{-11}$). After adjusting for these clinical variables with a multivariate Cox regression analysis, higher $PPS_{287}$ remained to be an independent prognostic factor for predicting overall survival (ZJCRC: HR = 3.24, $P = 1.05 \times 10^{-10}$; PLCO: HR = 2.25, $P = 2.72 \times 10^{-5}$) in the two cohorts.

To evaluate the additional prognostic value of $PPS_{287}$ to the traditional clinical model, we constructed a combined Cox regression

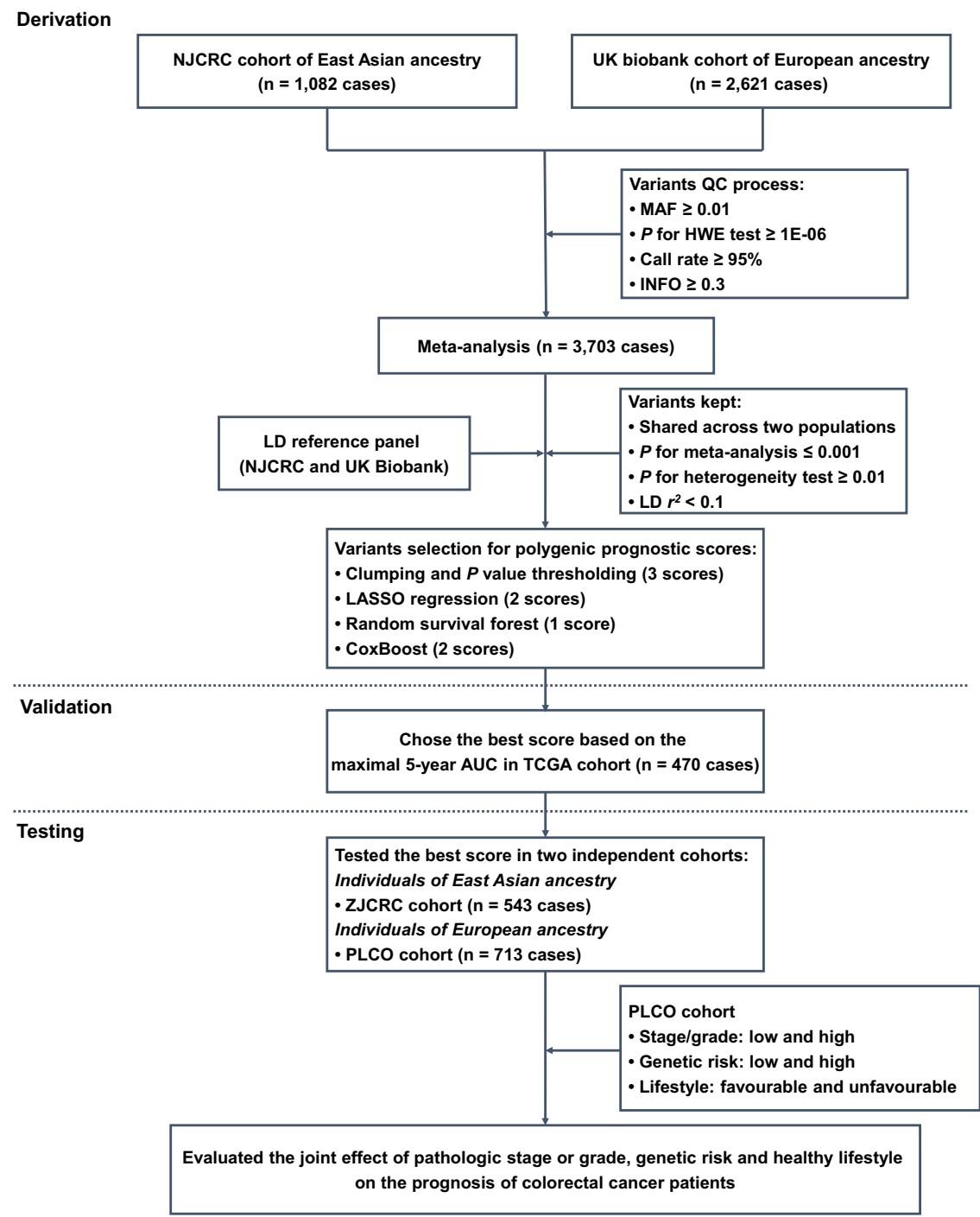

**Fig. 1 | Summary of the study design.** QC quality control, MAF minor allele frequency, HWE Hardy-Weinberg Equilibrium, LD linkage disequilibrium, LASSO least absolute shrinkage and selection operator, TCGA The Cancer Genome Atlas, AUC area under the curve, PLCO Prostate, Lung, Colorectal and Ovarian Cancer Screening Trial.

model by integrating $PPS_{287}$ with several common clinical factors for each cohort (ZJCRC: sex, age, smoking status and drinking status; PLCO: sex, age, smoking status, drinking status, stage and grade). Compared to the traditional model, the calibration curve of the combined model showed better agreement between the predicted and observed 5-year overall survival (Fig. 2B).

In addition, the AUCs at 5-year overall survival prediction of the traditional prognostic model were 0.644 in the ZJCRC cohort and 0.807 in the PLCO cohort, while those of the combined model were 0.699 and 0.834, respectively (Fig. 2C), indicating that the predictive accuracy of the combined prognostic model was significantly higher than that of the PPS or traditional models alone in the two cohorts

($P_{AUC} < 0.01$; Supplementary Table 8). Similar results were also observed using more evaluation metrics (e.g., Harrell's C index and Royston and Sauerbrei's $R^2_D$; Supplementary Table 9), as well as the decision curve analysis (DCA; Supplementary Fig. 5A, B), demonstrating the additional value of PPS in colorectal cancer survival prediction.

**Joint effects of pathologic characteristics, genetic risk and healthy lifestyle on overall survival of colorectal cancer**
Subsequently, given that the PLCO cohort included sufficient lifestyle information, we calculated an integrated healthy lifestyle score and aimed to evaluate the joint effect of pathologic stage or grade, genetic

**Table 1 | Basic characteristics of study subjects**

| Variable | Derivation stage | | Validation stage | Testing stage | |
|---|---|---|---|---|---|
| | NJCRC | UK Biobank | TCGA | ZJCRC | PLCO |
| Patients, N | 1082 | 2621 | 470 | 543 | 713 |
| Ancestry[a] | EAS | EUR | EUR | EAS | EUR |
| Median follow-up time (months) | 51.7 | 48.6 | 23.4 | 74.7 | 48.2 |
| Death, N (%) | | | | | |
| Yes | 340 (31.4%) | 779 (29.7%) | 100 (21.3%) | 152 (28.0%) | 177 (24.8%) |
| No | 742 (68.6%) | 1842 (70.3%) | 370 (78.7%) | 391 (72.0%) | 536 (75.2%) |
| Sex, N (%) | | | | | |
| Male | 647 (59.8%) | 1555 (59.3%) | 248 (52.8%) | 289 (53.2%) | 419 (58.8%) |
| Female | 435 (40.2%) | 1066 (40.7%) | 222 (47.2%) | 254 (46.8%) | 294 (41.2%) |
| Age (year), mean ± SD | 58.3 ± 12.7 | 65.2 ± 6.5 | 67.3 ± 12.5 | 63.5 ± 10.8 | 70.1 ± 6.6 |
| Smoking status, N (%) | | | | | |
| Ever | 303 (28.3%) | 1412 (54.0%) | – | 185 (34.2%) | 402 (56.4%) |
| Never | 768 (71.7%) | 1203 (46.0%) | – | 356 (65.8%) | 311 (43.6%) |
| Missing | 11 | 6 | 470 | 2 | 0 |
| Drinking status, N (%) | | | | | |
| Ever | 258 (24.2%) | 2539 (97.0%) | – | 137 (25.4%) | 577 (91.7%) |
| Never | 810 (75.8%) | 78 (3.0%) | – | 403 (74.6%) | 52 (8.3%) |
| Missing | 14 | 4 | 470 | 3 | 84 |
| Stage[b], N (%) | | | | | |
| 1 | 34 (3.2%) | – | 83 (18.4%) | – | 270 (38.2%) |
| 2 | 332 (31.1%) | – | 170 (37.7%) | – | 195 (27.6%) |
| 3 | 387 (36.2%) | – | 131 (29.0%) | – | 154 (21.8%) |
| 4 | 315 (29.5%) | – | 67 (14.9%) | – | 87 (12.3%) |
| Missing | 14 | 2621 | 19 | 543 | 7 |
| Grade[c], N (%) | | | | | |
| G1 | 38 (3.6%) | – | – | – | 63 (9.5%) |
| G2 | 785 (74.5%) | – | – | – | 471 (71.1%) |
| G3 | 230 (21.8%) | – | – | – | 119 (18%) |
| G4 | 0 | – | – | – | 9 (1.4%) |
| Missing | 29 | 2621 | 470 | 543 | 51 |

*TCGA* The Cancer Genome Atlas, *AJCC* American Joint Committee on Cancer, *PLCO* Prostate, Lung, Colorectal and Ovarian Cancer Screening Trial.
[a]*EAS* East Asian population, *EUR* European population.
[b]Dukes stage (stage A, stage B, stage C and stage D) for NJCRC cohort; AJCC stage (stage I, stage II, stage III and stage IV) for TCGA cohort; combined clinical and pathologic stage (stage I, stage II, stage III and stage IV) for PLCO cohort.
[c]*G1* well differentiated, *G2* moderately differentiated, *G3* poorly differentiated, *G4* undifferentiated.

risk and healthy lifestyle on the prognosis of colorectal cancer patients in the PLCO cohort (Supplementary Table 10). Broadly, there was a notable dose-response manner on decreasing overall survival probability in the pattern of higher stage/grade, higher genetic risk (higher PPS), and unfavorable lifestyle (lower lifestyle score) (log-rank $P = 4.86 \times 10^{-19}$; Fig. 3A), but no second-order multiplicative interaction between them was observed ($P_{interaction} = 0.145$). In particular, patients with a high stage/grade, a high genetic risk and an unfavorable lifestyle had a 27-fold increased risk of death than those with a low stage/grade, a low genetic risk and a favorable lifestyle (HR = 28.15, $P = 3.68 \times 10^{-9}$; Fig. 3B).

Interestingly, when stratifying patients by the categories of stage/grade and genetic risk, although few significant associations were observed, patients with colorectal cancer who maintained a healthy lifestyle could experience a lower risk of death (HR < 1; Table 3) than those who followed an unfavorable lifestyle. Especially, among patients with a low stage/grade and a low genetic risk, the overall survival rate ranged from 65.78% (unfavorable lifestyle) to 92.90% (favorable lifestyle; $P = 0.042$). Notably, among patients with a high stage/grade and a high genetic risk, the 5-year overall survival rate of those with an unfavorable lifestyle decreased to 41.9%, which could be increased to 49.52% among those with a favorable lifestyle (difference = 7.62%).

## Clinical application of the integrated prognostic model
To further apply the integrated model including clinical stage/grade, PPS287 and healthy lifestyle score in clinical practice, we developed a **Co**lo**R**ectal **C**ancer **S**urvival **P**rediction **S**ystem (CRC-SPS, http://njmu-edu.cn:3838/CRC-SPS/), including (i) "Colorectal cancer survival summary statistics" and (ii) "Colorectal cancer survival prediction" modules. The "About" page provides more details about the functions of this web server.

On the "Colorectal cancer survival summary statistics" page, when users enter a batch of SNP IDs, or enter a genetic region, a table [with chromosome ID, SNP ID, SNP genomic position, SNP alleles (A1: effect allele; A2: reference allele), effect allele frequency (EAF), beta, standard error (SE) in NJCRC and UK Biobank cohorts, and corresponding associations of meta-analysis] will be built. Users can download the results by clicking the "Download" button. Besides, users can select one SNP-survival pair and click the 'Plot' button, the diagrams of Kaplan–Meier plot will be provided to display the associations among the two cohorts.

**Table 2 | Performance of polygenic prognostic scores derived from different approaches in the TCGA cohort**

| Method | Parameter[a] | Weight[b] | $N_{SNP}$[c] | All follow-up HR (95% CI)[d] | P[d] | 3-year follow-up HR (95% CI)[d] | P[d] | AUC[e] | 5-year follow-up HR (95% CI)[d] | P[d] | AUC[e] |
|---|---|---|---|---|---|---|---|---|---|---|---|
| C + T | 0.001 | Meta | 300 | 1.97 (1.56, 2.50) | 1.70E−08 | 2.07 (1.56, 2.75) | 4.92E−07 | 0.653 | 2.04 (1.59, 2.61) | 1.82E−08 | 0.635 |
| C + T | 1.00E−04 | Meta | 34 | 1.47 (1.17, 1.85) | 0.001 | 1.45 (1.11, 1.91) | 0.007 | 0.606 | 1.45 (1.14, 1.84) | 0.002 | 0.541 |
| C + T | 1.00E−05 | Meta | 5 | 1.18 (0.96, 1.45) | 0.116 | 1.19 (0.94, 1.51) | 0.156 | 0.575 | 1.18 (0.95, 1.47) | 0.123 | 0.531 |
| LASSO | 0.01 | Meta | 287 | 1.95 (1.54, 2.46) | 2.41E−08 | 2.08 (1.57, 2.76) | 4.10E−07 | 0.635 | 2.04 (1.59, 2.61) | 1.63E−08 | 0.633 |
| LASSO | 0.01 | Penalty | 287 | 1.97 (1.55, 2.50) | 3.08E−08 | 2.10 (1.58, 2.77) | 2.15E−07 | 0.670 | 2.07 (1.61, 2.67) | 1.88E−08 | 0.643 |
| **RSF[f]** | **Optimal** | **Meta** | **287** | **1.99 (1.57, 2.52)** | **1.76E−08** | **2.06 (1.55, 2.75)** | **8.22E−07** | **0.659** | **2.05 (1.59, 2.63)** | **2.10E−08** | **0.652** |
| CoxBoost | Optimal | Meta | 265 | 1.97 (1.55, 2.49) | 2.27E−08 | 2.09 (1.58, 2.77) | 2.27E−07 | 0.657 | 2.07 (1.61, 2.66) | 1.02E−08 | 0.641 |
| CoxBoost | Optimal | Boosting | 265 | 1.64 (1.31, 2.06) | 1.75E−05 | 1.74 (1.33, 2.28) | 5.92E−05 | 0.665 | 1.67 (1.31, 2.12) | 2.80E−05 | 0.555 |

PPS polygenic prognostic score, TCGA The Cancer Genome Atlas, SNP single nucleotide polymorphism, C + T clumping and P value thresholding, LASSO least absolute shrinkage and selection operator, RSF random survival forest, HR hazard ratio, 95% CI 95% confidence interval, SD standard deviation, ROC receiver operating characteristics.
[a]Parameters for SNPs section: P value for C + T method; lambda for LASSO method; optimal AUC for RSF method; optimal boosting steps for CoxBoost method.
[b]Weight for PPS construction, derived from meta-analysis or penalized/boosted regression.
[c]Number of SNPs in the derivation stage.
[d]HR (95% CI) per SD, derived from cox regression model with the adjustment of sex, age, stage and top 10 principal components. The P value is two-sided.
[e]Area under the time-dependent ROC curve.
[f]The optimal PPS was highlighted in bold.

On the "Colorectal cancer survival prediction" page, CRC-SPS can help users estimate individual 5-year overall survival probability, with the PLCO cohort as a reference dataset. In brief, users can easily input their sex, age, lifestyle information (e.g., smoking status) and clinical characteristics (e.g., clinical stage) along with the genotypes of 287 SNPs to obtain an estimated 5-year survival probability. In addition, we provided the 5-year survival probability (i.e., 77.1%) in the PLCO cohort as a reference threshold, to stratify the population into subgroups with high and low risk of death. For example, the colorectal cancer patient with a predicted 65.8% of 5-year survival probability was grouped as having a high risk of death.

## Discussion

In the current study, we performed an EAS-EUR meta-analysis of colorectal cancer survival GWASs and found two suggestive genome-wide significant genetic loci (9p21.2 and 12q12) associated with colorectal cancer overall survival. Furthermore, we constructed and validated a robust PPS framework (PPS$_{287}$), independent of clinical factors, that could effectively stratify colorectal cancer survival in three independent longitudinal cohorts. Notably, the detrimental effect of pathologic characteristics and genetic risk on the prognosis of colorectal cancer could be attenuated by adherence to a healthy lifestyle.

Although previous GWASs have identified multiple SNPs associated with colorectal cancer risk, few studies have focused on the genetic architecture of survival outcomes[16–18]. For example, Wills et al. performed a survival GWAS among 1926 patients with advanced colorectal cancer, and supported rs79612564 (2q34) in *ERBB4* as a predictive biomarker of survival, as evidenced by the replication stage of independent colorectal cancer patients[17]. Here, leveraging the meta-analysis of EAS and EUR populations, we uncovered two variants, rs10967103 (9p21.2) and rs79067806 (12q12), linked to overall survival in colorectal cancer with substantial effect sizes (both HRs >1.5). Interestingly, these two prognostic variants were not associated with colorectal cancer susceptibility, indicating the diverse genetic background between the initiation and progression of colorectal cancer, which was consistent with previous findings[13,19]. Therefore, it will be necessary to identify variants carried with stronger effect sizes and increased statistical power among larger longitudinal populations, and to systematically decode the inconsistent features of the genetic architecture underlying the susceptibility and progression of colorectal cancer.

In recent decades, cumulative evidence has suggested the clinical utility of genetic biomarkers in estimating the risk of cancer death and improving patients' survival outcomes[5,20,21]. It is noteworthy that inherited germline variants (i.e., SNPs) are fixed at conception and do not change over time; therefore, they are considered as robust and cost-efficient biomarkers for personalized medicine. Currently, PRS, defined as a weighted sum of a set of risk-associated SNPs, has been demonstrated to be effective in identifying individuals at high risk of developing diseases[22,23]. For example, we ever developed a EAS-EUR PRS framework derived from genome-wide SNPs that can effectively predict colorectal cancer risk in EAS and EUR populations, indicating the potential application of PRS in colorectal cancer risk stratification[9]. However, there was no significant association between PRS and the increased risk of cancer mortality among cancer patients, as evidenced by several prospective studies[19,24] and our previous findings[13]. Therefore, considering the limited clinical utility of PRS in disease survival evaluation, we proposed a robust PPS$_{287}$ framework, independent of clinical factors, that could be used for colorectal cancer survival stratification in EAS and EUR populations, as evidenced by three independent cohorts. Notably, compared to low-PPS$_{287}$ patients, the subgroup with high PPS$_{287}$ showed poorer prognosis, and these patients could be recommended for colorectal cancer personalized therapy.

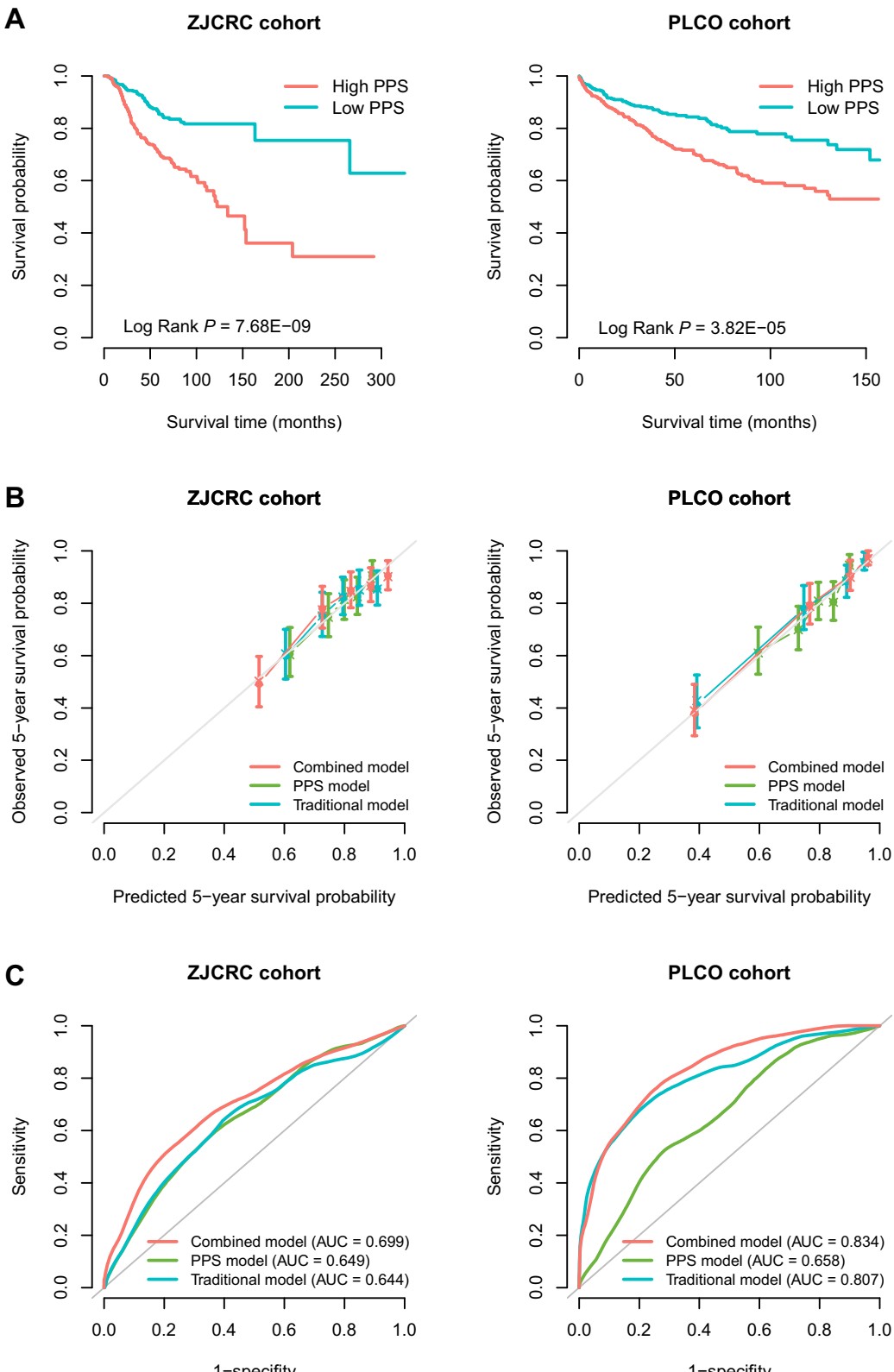

**Fig. 2 | Prognostic evaluation of the optimal polygenic prognostic score (i.e., PPS_287) in the ZJCRC and PLCO cohorts. A** Kaplan–Meier curves for overall survival probability stratified by different levels of PPS (based on median value) in the ZJCRC and PLCO cohorts. **B** Calibration curve of different prognostic models for predicting 5-year survival probability in the ZJCRC and PLCO cohorts. The vertical error bars denote the 95% CI. **C** Time-dependent ROC curves of different prognostic models regarding 5-year survival probability in the ZJCRC and PLCO cohorts. The traditional model included sex, age, smoking status and drinking status for the ZJCRC cohort; and sex, age, smoking status, drinking status, stage and grade for the PLCO cohort. The combined model included both traditional factors and PPS. The sample sizes of ZJCRC and PLCO cohorts are 543 and 713 cases. Note: PLCO Prostate, Lung, Colorectal and Ovarian Cancer Screening Trial, PPS polygenic prognostic score, ROC receiver operating characteristics, AUC area under the curve, 95% CI 95% confidence interval.

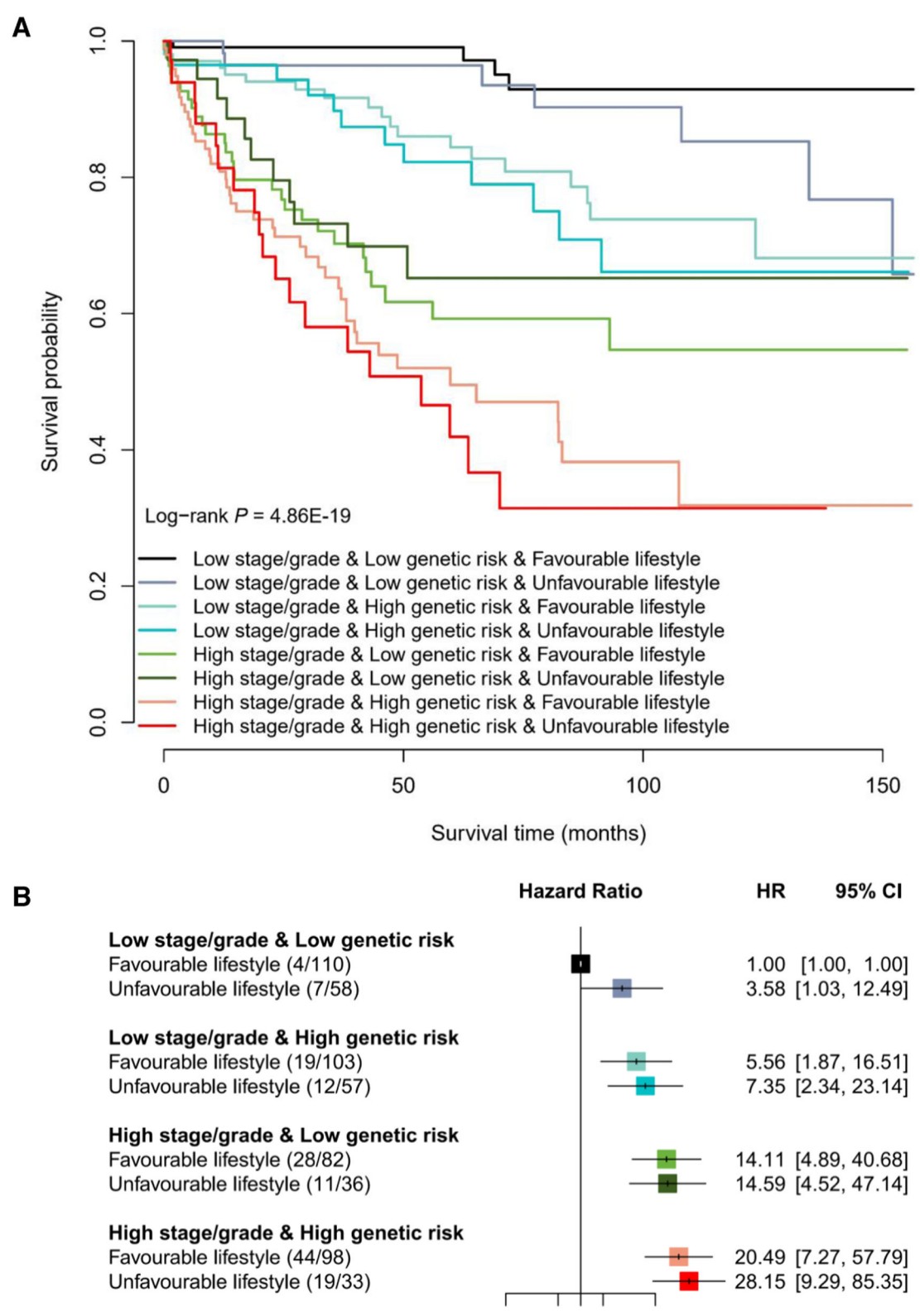

**Fig. 3 | The overall survival probability of colorectal cancer patients according to different levels of pathologic stage or grade, genetic risk, and healthy lifestyle in the PLCO cohort. A** Kaplan–Meier curves for overall survival probability stratified by different levels of pathologic stage or grade, genetic risk and healthy lifestyle. **B** The association of pathologic stage or grade, genetic risk and healthy lifestyle with overall survival of colorectal cancer patients. The HR and 95% CI were derived from the Cox regression model with the adjustment of sex, age, research center, arm and top 10 principal components. The number in the bracket indicates the number of deaths/number of all cases. The horizontal error bars denote the 95% CI. The sample size of PLCO cohort is 713 cases. Note: PLCO Prostate, Lung, Colorectal and Ovarian Cancer Screening Trial, HR hazard ratio, 95% CI 95% confidence interval.

**Table 3 | The association of pathologic stage or grade, genetic risk and healthy lifestyle with overall survival of colorectal cancer patients in the PLCO cohort**

| Stage/grade | Genetic risk | Lifestyle | Deaths/All[a] | OS[b] | 3-year OS[b] | 5-year OS[b] | HR (95% CI)[c] | P[c] |
|---|---|---|---|---|---|---|---|---|
| Low | Low | Favorable | 4/110 | 92.90% | 99.08% | 99.08% | 0.17 (0.03, 0.94) | 0.042 |
| | | Unfavorable | 7/58 | 65.78% | 96.43% | 96.43% | 1.00 (reference) | |
| | High | Favorable | 19/103 | 68.16% | 91.66% | 84.42% | 0.51 (0.21, 1.21) | 0.124 |
| | | Unfavorable | 12/57 | 66.12% | 89.75% | 82.25% | 1.00 (reference) | |
| High | Low | Favorable | 28/82 | 54.69% | 70.26% | 59.24% | 0.95 (0.38, 2.33) | 0.904 |
| | | Unfavorable | 11/36 | 65.21% | 73.19% | 65.21% | 1.00 (reference) | |
| | High | Favorable | 44/98 | 31.85% | 65.30% | 49.52% | 0.78 (0.40, 1.54) | 0.477 |
| | | Unfavorable | 19/33 | 31.42% | 58.04% | 41.90% | 1.00 (reference) | |

*PLCO* Prostate, Lung, Colorectal and Ovarian Cancer Screening Trial, *HR* hazard ratio, *95% CI* 95% confidence interval.
[a]Number of deaths/number of all cases.
[b]OS, overall survival probability.
[c]Derived from the Cox regression model with the adjustment of sex, age, research center, arm and top 10 principal components. The P value is two-sided.

Importantly, by integrating different categories of pathologic characteristics (i.e., clinical stage or grade), genetic risk and healthy lifestyle, we developed an analytical framework for colorectal cancer survival stratification. Interestingly, adherence to a healthy lifestyle could attenuate the risk of death, especially evident among patients with low stage/grade and low genetic risk (*P* < 0.05). Notably, among patients with a high stage/grade and a high genetic risk, the 5-year overall survival rate of an unfavorable lifestyle could be increased by 7.62% with adherence to a favorable lifestyle, further emphasizing the public notion that a healthy lifestyle among colorectal cancer patients can lead to an evident reduction in death[14,15].

Our study has several strengths. First, we performed a EAS-EUR meta-analysis of colorectal cancer survival GWASs and identified two significant variants associated with overall survival of colorectal cancer. Second, we proposed and validated a robust PPS framework that could be effectively used for colorectal cancer survival stratification among EAS and EUR populations. Third, leveraging the information of pathologic characteristics, genetic risk and lifestyle, we developed a user-friendly web server to generate a customized estimate of 5-year survival probability for colorectal cancer patients, for use as a potential tool in personalized survival prediction. Nevertheless, we also need to acknowledge some limitations. First, we only included a total of 3703 colorectal cancer patients (i.e., NJCRC and UK Biobank cohorts) for the survival-based meta-analysis, with the limitation of statistical power for detecting genome-wide significant loci; thus, more datasets should be included when available in the future. Second, clinical stage and grade, as important prognostic factors, are not available in some cohorts (i.e., UK Biobank and ZJCRC), which should be further included for survival evaluation; besides, additional survival outcome-related factors (e.g., treatment) are also needed to be considered. Third, the lifestyle or other confounding factors were derived from the baseline questionnaire in the PLCO cohort, which could not reflect the dynamic changes during the follow-up after colorectal cancer diagnosis; thus, more detailed surveillance is also needed. Fourth, only EAS and EUR ancestry groups were included for PPS construction, other ethnic groups (e.g., African Americans and Hispanics), as well as more sophisticated methods should be considered in the future work. In addition, the model performance and benefit of healthy lifestyle maintenance need to be further validated using a larger longitudinal population with sufficient follow-up time and sample size.

In conclusion, leveraging the colorectal cancer survival GWAS meta-analysis and multi-center cohorts, we constructed and validated a robust PPS framework that could effectively predict colorectal cancer survival among EAS and EUR populations. Importantly, we also provided further evidence that a healthy lifestyle could attenuate the detrimental impact of pathologic characteristics and genetic risk on colorectal cancer progression, which could shed additional light on precision clinical management of colorectal cancer.

## Methods
### Study subjects
#### Derivation stage
**NJCRC cohort of EAS ancestry.** The subjects in the NJCRC cohort were recruited from the National ColoRectal Cancer Cohort (NCRCC), including 1082 Chinese patients, being part of the Genetics and Epidemiology of Colorectal Cancer Consortium (GECCO). Detailed information can be found in the Supplementary Methods[9,25].

**UK Biobank cohort of EUR ancestry.** The UK Biobank cohort (https://www.ukbiobank.ac.uk/) is a prospective, population-based study that recruited 502,528 adults aged 40–69 years from the general population between April 2006 and December 2010[26]. After applying individual-level filtering criteria (Supplementary Methods), a total of 2621 incident colorectal cancer cases of EUR ancestry were retained for our analysis[27]. This study was conducted using the UK Biobank Resource under Application #45611.

#### Validation stage
**TCGA cohort of EUR ancestry.** TCGA (https://www.cancer.gov/about-nci/organization/ccg/research/structural-genomics/tcga) is a joint cancer genomics program of the National Cancer Institute and National Human Genome Research Institute that began in 2006[28]. Over the past decade, TCGA has collected more than 20,000 primary cancer and matched normal samples from over 10,000 cases across 33 cancer types. Here, a total of 470 individuals of EUR ancestry with colorectal cancer were retained for further analysis[13].

#### Testing stage
**ZJCRC cohort of EAS ancestry.** The 543 Chinese colorectal cancer cases in the ZJCRC cohort were derived from the Jiashan Institute of Cancer Prevention and Treatment. The population details were described in the Supplementary Methods[9].

**PLCO cohort of EUR ancestry.** The PLCO cancer screening trial is a cohort study that aims to evaluate the accuracy and reliability of screening methods for prostate, lung, colorectal, and ovarian cancer[29]. Based on the filtering criteria, a total of 713 white individuals of EUR ancestry with colorectal cancer remained in the subsequent analysis. Detailed information was described in the Supplementary Methods[30]. This study was approved by the ethics committees of the PLCO consortium providers (#PLCO-84).

The basic information of each cohort has been described in the Table 1, and the distribution of genetic ancestry is shown in the

Supplementary Fig. 1. All participants provided written informed consent prior to data collection. Our study was approved by the ethics committee of Nanjing Medical University.

### Genotyping, imputation and quality control (QC)

For each cohort, the detailed information about genotyping and imputation process is described in the Supplementary Methods. Subsequently, the imputed SNPs located in autosomal chromosomes were removed if they had (i) minor allele frequency (MAF) < 0.01; (ii) call rate <95%; (iii) Hardy-Weinberg equilibrium (HWE) $P$ value $< 1 \times 10^{-6}$ and (iv) information metric (info score) <0.3.

### Definition of overall survival

The follow-up time of overall survival was calculated from the date of colorectal cancer diagnosis to the date of death from any cause or the end of the follow-up period for censoring.

### Meta-analysis of colorectal cancer survival GWAS

We used the Cox proportional hazards model to calculate HR and 95% confidence interval (CI) for the association between each SNP and colorectal cancer survival, separately for the NJCRC and UK Biobank cohorts, with the adjustment of corresponding covariates [NJCRC: sex, age, smoking status, drinking status, grade, stage and first 10 principal components; UK Biobank: sex, age, body mass index (BMI), smoking status, drinking status and first 10 principal components].

Furthermore, leveraging the summary statistics of the two survival GWASs (totally 3703 cases), a meta-analysis in an inverse variance-weighted fixed-effects model was performed to identify survival-associated variants across EAS and EUR ancestries, implemented by METAL software[31]. We then retained SNPs for subsequent analysis if they (i) passed filters in both the EAS (i.e., NJCRC cohort) and EUR (i.e., UK Biobank cohort) populations; (ii) did not show substantial heterogeneity among studies ($P$ value for heterogeneity test ≥0.01); and (iii) harbored a significant association with colorectal cancer survival ($P$ value for meta-analysis ≤0.001). Finally, also considering that the consistency of SNPs in at least one external dataset, a total of 300 independent SNPs (linkage disequilibrium, LD $r^2 < 0.1$) were kept, and variants at $P$ value $< 5 \times 10^{-6}$ were considered to be suggestively genome-wide significant.

In addition, we applied a colorectal cancer GWAS meta-analysis of case-control studies to evaluate the risk effect of genome-wide significant prognostic variants[9]. The meta-analysis was performed with totally 35,145 cases and 288,934 controls of EAS and EUR ancestries, derived from NJCRC (1316 cases and 2207 controls; EAS), BJCRC (932 cases and 966 controls; EAS), SHCRC (1116 cases and 1054 controls; EAS), ZJCRC (1046 cases and 1184 controls; EAS), BioBank Japan Project (BBJ; 7062 cases and 195,745 controls; EAS), GECCO (21,608 cases and 20,278 controls; EUR) and PLCO (2065 cases and 67,500 controls; EUR) GWASs.

### Calculation of PPS

To aggregate the weak effect of individual SNPs, we calculated PPS using the following formula: $PPS = \sum_{i=1}^{n} \beta_i SNP_i$, where $n$ is the number of selected SNPs, $SNP_i$ and $\beta_i$ are the number of effect alleles (i.e., 0, 1, 2) and weight corresponding to the $i$-th SNP, respectively. Using the genotype data of 300 independent SNPs, we constructed eight candidate PPSs for colorectal cancer survival prediction through four approaches, including classic clumping and $P$ value thresholding[32] (i.e., C + T, 3 scores), LASSO[33] (2 scores), RSF[34] (1 score), and CoxBoost[35] (2 scores) methods. The details are described in the Supplementary Methods.

### Calculation of healthy lifestyle score

The construction of healthy lifestyle score was based on our previous study[9], of which included common lifestyle factors, and we kept lifestyle factors with low missing rate for analysis. Briefly, we calculated healthy lifestyle scores based on five common lifestyle factors in the PLCO cohort, derived from the baseline questionnaire and diet history questionnaire (DHQ), including BMI, tobacco smoking, alcohol consumption, red and processed meat intake, and vegetable and fruit intake. Each lifestyle factor was given a score of 0 or 1, with 1 representing the healthy behavior category, and the sum of the five scores was used as the healthy lifestyle score. The detailed information is shown in the Supplementary Table 1.

### Statistical analysis

The Manhattan plot and quantile-quantile plot based on the -$\log_{10}$ ($P$ value) were created by using R package $qqman$. The heterogeneity was measured using Cochran's Q statistics and $I^2$.

In the validation (i.e., TCGA) and testing (i.e., ZJCRC and PLCO) cohorts, we used the Cox proportional hazards model to estimate the HRs and 95% CIs for the association of PPS with colorectal cancer survival after adjusting for corresponding confounding factors. All datasets were analyzed underlying complete case analysis. The discriminatory ability of the prognostic model (i.e., Cox regression model) was evaluated using the time-dependent ROC curve [the optimal estimation of sensitivity and specificity was based on the Index of Union (IU) method[36]] using R package $survivalROC$, with a bootstrap method of 10,000 iterations for calculating 95% CI and ROC comparison. In addition, the Harrell's C index and Royston and Sauerbrei's $R^2_D$ in Cox proportional hazards models were also used for evaluating model performance[37]. The DCA plot was also used to demonstrate the clinical benefit of different models at 5 years of follow-up, using R package $dcurves$. Participants were then classified into two genetic-risk subgroups (including low-PPS and high-PPS) according to the median value of PPS for group comparison. The Kaplan–Meier curve and log-rank test were used to evaluate the difference in overall survival probability stratified by different levels of PPS. In addition, to assess the robustness of the PPS in survival prediction, we performed the following sensitivity analyses: (i) excluded colorectal cancer patients who died during the first year of follow-up; (ii) evaluated the associations using ancestry-corrected PPS (briefly, fit a linear regression model using the first ten principal components of ancestry to predict PPS, and the residual from this model was used to create ancestry-corrected PPS)[9].

In the PLCO cohort, participants were further classified into low stage/grade [i.e., low stage (stage I and stage II) and low grade (G1 and G2)] and high stage/grade [i.e., high stage (stage III and stage IV) or high grade (G3 and G4)] subgroups, as well as unfavorable (i.e., 0 and 1 lifestyle score) and favorable (i.e., ≥ 2 lifestyle score) subgroups. The log-rank test and Cox proportional hazards model were used to evaluate the association of different levels of pathologic stage/grade, genetic risk or healthy lifestyle with overall survival probability of colorectal cancer. The R package $Shiny$ was used to construct the colorectal cancer survival prediction web server, which was freely available and open source.

All statistical analyses were performed using R software (version 4.0.3), and a two-sided $P$ value less than 0.05 indicated statistical significance.

### Reporting summary

Further information on research design is available in the Nature Portfolio Reporting Summary linked to this article.

## Data availability

The raw genotype and clinical data of European populations have been deposited in UK Biobank (https://www.ukbiobank.ac.uk/; Application #45611), TCGA [https://www.cancer.gov/about-nci/organization/ccg/research/structural-genomics/tcga, available on the database of Genotypes and Phenotypes (dbGaP) accession: phs000178.v11.p8) and

PLCO (https://dceg.cancer.gov/research/who-we-study/cohorts/prostate-lung-colon-ovary-prospective-study; Application #PLCO-84, available on the dbGaP accessions: phs001286.v1.p1, phs001415.v1.p1, phs001078.v1.p1 and phs001554.v1.p1) programs. The data of Chinese populations have been deposited into Open Archive for Miscellaneous Data (OMIX) of the National Genomics Data Center of China (BioProject ID: PRJCA023932), which can be shared upon academic request to the corresponding author (M.W., mwang@njmu.edu.cn) in accordance with the Chinese genomic data sharing policy, with about three months for data preparation and one year for data using. The summary statistics of meta-analysis and detailed information for $PPS_{287}$ calculation are provided in CRC-SPS. The $PPS_{287}$ weight files are also available in PGS Catalog (https://www.pgscatalog.org/; PGS ID: PGS004586).

## Code availability
For genotype imputation processing, SHAPEIT and IMPUTE2 (https://mathgen.stats.ox.ac.uk/impute/impute_v2.html) were used. R (version 4.0.3, https://www.r-project.org/) was used for the development and validation of PPS, the details have been described in the Supplementary Methods.

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

## Acknowledgements

We thank The Cancer Genome Atlas (TCGA), Prostate, Lung, Colorectal and Ovarian (PLCO) cancer screening trial (Application #PLCO-84) and UK Biobank cohort (Application #45611) for sharing colorectal cancer GWAS data. This study is funded by the National Natural Science Foundation of China (81822039, M.W.; 82073631, D.G.).

## Author contributions

M.W., M.D. and H.S. supervised the entire project. M.W., J.X., M.D., D.G. and S.L. contributed to the data interpretation, data analysis, and writing of the draft. S.Q., Y.C., W.S., S.B., S.C., L.Z., M.J., K.C., Z.H. and Z.Z. contributed to the study design, sample collection, and experiment or data interpretation. All authors reviewed or revised the manuscript and approved the final draft for submission.

## Competing interests

The authors declare no competing interests.
