## [Peer Review File · Nature Communications]

REVIEWER COMMENTS

Reviewer #1 (Remarks to the Author):

Xin and colleagues presented a large-scale genome-wide association analysis of survival among colorectal cancer patients. First, they combined two independent cohorts (UK Biobank and NJCRC) representing European and East Asian ancestry in the meta-analysis and discovered two novel survival-associated loci. Furthermore, they constructed several survival-based polygenic prognostic scores (PPSs), and determined an optimal PPS in a validation cohort. Importantly, the optimal PPS was replicated in two external cohorts of different ancestry background.

In total, this is an interesting and innovative study that firstly introduces PPS (a germline variant score) into the colorectal cancer survival prediction. Here, I have several suggestions that could further improve the study as follows:

1. This study included five cohorts, derived from different ethnic groups (European and East Asian). Although most cohorts have been described in previous studies, the authors should provide the principal component analysis (PCA) plot for showing the ancestry background for each cohort in the Supplementary files.
2. The authors used the imputation information (INFO) metric of 0.3 as a cut-off for quality control, which is a common but not strict threshold. Please check the INFO of 287 variants in PPS construction and ensure them are reliable for future clinical application.
3. The authors proposed a new survival evaluation framework, not only including common clinical stage or grade, but also genetic risk and healthy lifestyle. This is important for colorectal cancer patients' individualized therapy. Here, the lifestyle factors in the PLCO were from baseline questionnaire, not after the diagnosis, which should be further reflected in the discussion.
4. I thank the authors that provided an online web-server, CRC-SPS, for helping users estimate the 5-year survival. To improve the utility of the server, I suggest (i) add a module "Survival GWAS summary statistics", that shows the associations of each SNP with colorectal cancer survival in the discovery stage of UK Biobank and NJCRC cohorts; (ii) add an example plot and more details in the About page, to help users better understand your estimated results.
5. The construction of PPS₂₈₇ was valuable for the clinical application, please provide more details (e.g., allele frequency, effect size in different ancestries and heterogeneity statistics) in the Supplementary Tables or CRC-SPS webserver About page.

Minor:

1. It may be interesting to investigate the association between PPS and some population characteristics (e.g., sex, age).
2. The authors used a previous case-control GWAS (35145 cases and 288934 controls) to evaluate the risk effect of two novel survival-associated loci, please provide more population details in the Method.

Reviewer #2 (Remarks to the Author):

This paper reports the development of a polygenic predictive score for bowel cancer survival. Survival GWAS have been performed with two datasets (NJCRC and UKB, Chinese and European ancestry respectively) with subsequent meta-analysis, identifying two loci with suggestive associations, though neither had clear functional associations. PPS were then developed in the same cohort, and performance compared in a validation cohort of European individuals (TCGA). The best performing PRS was then tested in cohorts of Chinese (ZJCRC) and European (PLCO) ancestry, and combined with a healthy .

Overall, although a clearly planned methodology and results are presented, there are significant methodological limitations and an overly-optimistic interpretation of the strength of the findings and their impact on clinical practice, which is not supported by the data presented. In addition, important components of the methodology are absent from the paper, as detailed below. Addressing these issues would significantly strengthen the paper.

I would suggest that that number of cases included does not constitute “Large scale” GWAS meta-analysis. As noted, there is insufficient power to detect GWAS-significant SNP associations. This is an inevitable limit of the available datasets which should be made clearer in the discussion. Is there a reason that existing prognostic GWAS analysis (e.g. Wills et al noted in the discussion) was not included in the meta-analysis? Was this data unavailable, or overlap between datasets?

A significant methodological issue in the construction of the PPS is the use of the same dataset for both GWAS and PPS parameterisation. This overlap will result in overfitting of PPS (Wray NR, Yang J, Hayes BJ, Price AL, Goddard ME, Visscher PM. Pitfalls of predicting complex traits from SNPs. *Nat Rev Genet* 2013;14:507-15. doi:10.1038/nrg3457) and optimism in performance estimates. This is clearly demonstrated by the drop in performance of the top performing PRS shown in the TCGA cohort in which calibration and AUC are quite poor (Supp figure 3) compared to the NJCRC and UKB cohorts (Supp fig 2).

I found the use of both a ‘healthy lifestyle score’ (HLS) and ‘traditional model’ in the PLCA cohort confusing. The HLS is constructed as a categorised risk counting model, and is used in the evaluations of association between pathological features/genetic risk and lifestyle. In the construction of the HLS - it is unclear why these particular predictors were chosen (based on existing literature?), and how were cut off thresholds defined. Separately, the ‘traditional model’ includes some but not all of these lifestyle predictors and this is used as the comparator and base for the combined model. The authors reference a pre-existing lifestyle prediction model for overall survival in bowel cancer patients which has been externally validated (Cheng et al) – is there a reason that this was not used as the traditional model?

Handling of missing data noted in Table 1 and several supplementary tables is not reported – was complete case analysis used, or imputation in some form?

The methodology for the construction of the combined model was not clearly defined in the paper. Line 174 – “we constructed a combined model by integrating PPS287 with several clinical

factors for each cohort” is unclear. I presume this was in a Cox-regression analysis? If so how were the PRS and traditional risk predictors combined?

Reported statistics are limited to effect sizes (HR), AUC, and KM curves/log rank tests. Reporting of other key metrics such as variance explained for the PPS would be helpful, in addition to more clinically meaningful metrics such as sensitivity/specificity.

Rather than the results of traditional vs combined model performance in the ZJCRC and PLCO cohort showing that the PRS improves performance, I would argue that these two cohort demonstrate that overwhelmingly the presence of stage and grade are the strongest predictors of survival (AUC of 0.8 compared to 0.63 – Figure 2B/C and Supp Table 8). In Figure 2 – whilst evaluation of calibration of the PRS in the ZJCRC and PLCO cohorts is helpful, the ‘traditional’ and ‘combined’ models in these cohorts ought to be near perfect, as these have been developed in these datasets. Whether the addition of PRS to a model containing stage and grade is of any value needs much more careful consideration than the reporting of the increment in AUC. As above, reporting of variance explained would be helpful, but more meaningfully, clinically relevant metrics such as sensitivity/specificity/PPV, and decision curve analysis, would help elucidate what the incremental gain of PRS would be. See Wand et al <https://doi.org/10.1038/s41586-021-03243-6> for reporting standards for polygenic scores in risk prediction studies.

From the data in Table 3 and Figure 3, although the data show an improvement in OS with healthy lifestyles for some categories this is minimal (e.g. high stg/grade and low genetic risk) and I would argue that the results suggest lifestyle has a more marked impact on risk if stage/grade and genetic risk are low, rather than the discussion of the major effect in higher risk groups (discussion Line 267). I note that the 5 year survival is referenced, whilst the median follow-up for the cohort is 4 years which may limit the strength of this conclusion. Potential confounding from changes in lifestyle over the period of follow-up (for example, prompted by a diagnosis of higher stage/grade cancer) should be considered.

The combined model has been made publicly available for use ‘in clinical practice’ and is proposed as an important tool for personalised prediction (Line 278). This model has not been externally validated, having been constructed and tested here only in the PLCO cohort, and I have very serious reservations about it being made available as such as a clinical tool. I would welcome its availability for further research. The appropriateness of the PLCO cohort as the reference dataset for a general CRC population as a reference threshold for high/low risk of death is also not considered.

Some more minor comments:

Line 72 The sentence “However, survival probability...” Doesn’t quite make sense to me

The paper does not report how cancer cases were identified in the cohorts studied (e.g. which field from UKB was used to ID the cases – first reported instance, or registry data?) End of follow-up/censoring dates should also be defined, as cancer data has been regularly updated for UKB. This could be inserted in the supplementary methods.

Line 131 – C+T, RSF etc should be defined when first mentioned.

Response to the Reviewers' comments:

We appreciate the reviewers for the insightful suggestions. They helped us improve the quality of our manuscript. Here are the point-by-point responses to the comments and concerns.

Reviewer #1 (Remarks to the Author):

Xin and colleagues presented a large-scale genome-wide association analysis of survival among colorectal cancer patients. First, they combined two independent cohorts (UK Biobank and NJCRC) representing European and East Asian ancestry in the meta-analysis and discovered two novel survival-associated loci. Furthermore, they constructed several survival-based polygenic prognostic scores (PPSs), and determined an optimal PPS in a validation cohort. Importantly, the optimal PPS was replicated in two external cohorts of different ancestry background.

In total, this is an interesting and innovative study that firstly introduces PPS (a germline variant score) into the colorectal cancer survival prediction. Here, I have several suggestions that could further improve the study as follows:

Response: We thank Reviewer #1 for the summary and positive comments.

- This study included five cohorts, derived from different ethnic groups (European and East Asian). Although most cohorts have been described in previous studies, the authors should provide the principal component analysis (PCA) plot for showing the ancestry background for each cohort in the Supplementary files.*

Response: Thank you for the suggestions. We have added the principal component analysis (PCA) for the five cohorts, including the NJCRC and UK Biobank cohorts of discovery stage, TCGA cohort of validation stage, ZJCRC and PLCO cohorts of testing stage (**Figure R1**). Through integration with the 1000 Genomes Project populations, we could observe that the NJCRC and ZJCRC cohorts were included in East Asian populations, while UK Biobank, TCGA and PLCO were limited to European populations, which has been added as **Supplementary Figure 1** in the manuscript.

Figure R1 (Supplementary Figure 1). Principal component analysis (PCA) for the five cohorts (discovery stage: NJCRC and UK Biobank cohorts; validation stage: TCGA cohort; testing stage: ZJCRC and PLCO cohorts) of colorectal cancer patients and 1000 Genomes Project populations.

Note: CHB, Han Chinese in Beijing, China; JPT, Japanese in Tokyo, Japan; CEU, Utah residents with Northern and Western European ancestry; YRI, Yoruba in Ibadan, Nigeria; TCGA, The Cancer Genome Atlas; PLCO, Prostate, Lung, Colorectal and Ovarian Cancer Screening Trial.

2. *The authors used the imputation information (INFO) metric of 0.3 as a cut-off for quality control, which is a common but not strict threshold. Please check the INFO of 287 variants in PPS construction and ensure them are reliable for future clinical application.*

Response: Thank you for the valuable comments. We have checked the INFO value for each variant used in PPS construction (**Table R1**), and found that the INFO ranged from 0.72 to 1.00 in the NJCRC and UK Biobank cohorts, indicating that the imputation performance of 287 variants was overall adequate. The results have also been added in the **Supplementary Table 4**.

3. *The authors proposed a new survival evaluation framework, not only including common clinical stage or grade, but also genetic risk and healthy lifestyle. This is important for colorectal cancer patients' individualized therapy. Here, the lifestyle factors in the PLCO were from baseline questionnaire, not after the diagnosis, which should be further reflected in the discussion.*

Response: We appreciate the reviewer for pointing it out. We have added the limitation about lifestyle in the Discussion as follows:

“Third, the lifestyle or other confounding factors were derived from the baseline questionnaire in the PLCO cohort, which could not reflect the dynamic changes during the follow-up after colorectal cancer diagnosis; thus, more detailed surveillance is also needed.” in page 13.

Table R1 (Supplementary Table 4). Summary of the 287 variants for polygenic prognostic score construction.

CHR	SNP	Position ^a	Allele ^b	NJCRC			UK Biobank				Meta-analysis			
				EAF ^c	HR (95% CI) ^d	P ^d	INFO ^e	EAF ^c	HR (95% CI) ^f	P ^f	INFO ^e	HR (95% CI) ^g	P ^g	P _{het} ^h
1	rs6668802	10924043	A/G	0.014	1.99 (1.07, 3.69)	2.86E-02	0.864	0.047	1.34 (1.09, 1.66)	5.97E-03	1.000	1.4 (1.15, 1.71)	9.36E-04	0.237
1	rs77424575	59846569	C/G	0.933	1.04 (0.74, 1.45)	8.37E-01	0.961	0.835	1.33 (1.15, 1.54)	1.13E-04	1.000	1.28 (1.12, 1.46)	2.92E-04	0.177
1	rs11209063	67855142	T/G	0.742	1.24 (1.02, 1.5)	2.84E-02	1.000	0.885	1.29 (1.09, 1.53)	3.53E-03	1.000	1.27 (1.12, 1.44)	2.77E-04	0.763
1	rs1415068	87055530	A/G	0.643	1.2 (1.01, 1.42)	3.37E-02	0.987	0.896	1.29 (1.07, 1.54)	6.29E-03	1.000	1.24 (1.1, 1.4)	6.30E-04	0.591
1	rs12737643	38486480	T/C	0.949	0.82 (0.58, 1.15)	2.45E-01	0.998	0.878	0.79 (0.68, 0.91)	9.28E-04	1.000	0.79 (0.69, 0.9)	4.58E-04	0.844
1	rs79585663	863138	A/C	0.651	0.78 (0.67, 0.91)	1.76E-03	0.992	0.988	0.7 (0.46, 1.05)	8.15E-02	1.000	0.77 (0.66, 0.89)	3.94E-04	0.609
1	rs78033786	218828298	T/C	0.057	1.39 (1.02, 1.9)	3.54E-02	0.998	0.042	1.44 (1.15, 1.8)	1.33E-03	1.000	1.42 (1.19, 1.71)	1.26E-04	0.867
1	rs1327609	81755444	T/C	0.049	0.69 (0.46, 1.04)	7.37E-02	1.000	0.423	0.86 (0.77, 0.95)	3.04E-03	1.000	0.84 (0.76, 0.93)	9.22E-04	0.316

.....

^aChromosomal position, hg19/GRCh37 build.

^bEffect/reference allele.

^cEAF, Effect allele frequency.

^dDerived from Cox regression model with the adjustment of sex, age, smoking status, drinking status, stage, grade and top 10 principal components in the NJCRC cohort.

^eImputation INFO value.

^fDerived from Cox regression model with the adjustment of sex, age, smoking status, drinking status and top 10 principal components in the UK Biobank cohort.

^gMeta-analysis between NJCRC and UK Biobank cohorts of East Asian and European populations.

^h*P* value for heterogeneity test.

4. I thank the authors that provided an online web-server, CRC-SPS, for helping users estimate the 5-year survival. To improve the utility of the server, I suggest (i) add a module "Survival GWAS summary statistics", that shows the associations of each SNP with colorectal cancer survival in the discovery stage of UK Biobank and NJCRC cohorts; (ii) add an example plot and more details in the About page, to help users better understand your estimated results.

Response: We appreciate the reviewer for the insightful suggestion. Here, we have further improved the CRC-SPS web server (<http://njmu-edu.cn:3838/CRC-SPS/>), with two functions: "Colorectal cancer survival summary statistics" and "Colorectal cancer survival prediction".

On the "Colorectal cancer survival summary statistics" page, when users enter a batch of SNP IDs, or enter a genetic region, a table (with chromosome ID, SNP ID, SNP genomic position, SNP alleles (A1: effect allele; A2: reference allele), effect allele frequency (EAF), beta, standard error (SE) in NJCRC and UK Biobank cohorts, and corresponding associations of meta-analysis) will be built (**Figure R2**). Users can download the results by clicking the "Download" button.

Show 10 entries Search:

CHR	SNP	BP_hg19	A1	A2	EAF_NJCRC	beta_NJCRC	se_NJCRC	EAF_UKBB	beta_UKBB	se_UKBB	beta_meta	se_meta	P_meta	Direction	HetSq	HetPVal
1	rs16038810	4423578	A	G	0.749	0.0384051713784195	0.0906323491597814	0.92624	-0.0032099859521992	0.0989582521295066	0.0194	0.0668	0.7714	+-	0	0.7565
1	rs6424059	3773106	T	C	0.319	0.0377699696361849	0.0838421219101327	0.4346	-0.0619427004948542	0.0506538964216451	-0.0353	0.0434	0.4158	+-	3.5	0.3087
1	rs12406076	2379550	C	G	0.609	0.0779004899200751	0.0816161272707819	0.8083	-0.0763995597081146	0.0631899655342319	-0.0186	0.05	0.7101	+-	55.3	0.1349
1	rs120034794	2937551	T	G	0.5639	-0.0960196248721509	0.0789669525513623	0.81	0.0945687750168061	0.0673769791290158	0.0143	0.0513	0.7806	+-	70.3	0.06635
1	rs12123665	4222442	T	C	0.04935	0.295762664419302	0.163899638997344	0.225	0.00106815732261193	0.0626532215975035	0.0386	0.0585	0.5091	++	64.5	0.09306
1	rs12128330	4092926	A	G	0.01198	-0.251484379053544	0.384105734908778	0.131	0.0358444859883565	0.0762462315526737	0.025	0.0748	0.7387	+-	0	0.4631
1	rs78458484	4426359	A	C	0.1335	0.061035840469226	0.114583884742438	0.013	-0.235857077189045	0.245957704452015	0.0061	0.1039	0.9379	+-	16.5	0.2739
1	rs2455103	3178416	A	G	0.7794	0.102664686775918	0.09892656082050218	0.8574	-0.00600593256736726	0.0723884243428679	0.0319	0.0584	0.5852	+-	0	0.3753
1	rs10915611	4872651	T	C	0.1084	-0.0108783184960645	0.128852712796532	0.8057	0.0662804961239953	0.0662396293032068	0.0502	0.0589	0.3946	+-	0	0.5943
1	rs72633383	3412726	T	C	0.04937	-0.323578625123182	0.208730505746887	0.02923	-0.126534176882851	0.161452112246855	-0.2003	0.1277	0.1168	--	0	0.4552

Showing 1 to 10 of 5,570 entries Previous 1 2 3 4 5 ... 575 Next

Figure R2. Example of the summary statistics on the "Colorectal cancer survival summary statistics" page.

In addition, users can select one SNP-survival pair and click the 'Plot' button, and the diagrams of Kaplan-Meier (KM) plot will be provided to display the associations among NJCRC (East Asian ancestry) and UK Biobank (European ancestry) cohorts. For example, patients with the SNP rs2742681 GT or TT genotypes have better survival than patients with rs2742681 GG genotype (P for log-rank test < 0.05 ; **Figure R3**).

Figure R3. Example of the association of rs2742681 with colorectal cancer overall survival in the NJCRC and UK Biobank cohorts on the “Colorectal cancer survival summary statistics” page.

On the “Colorectal cancer survival prediction” page, when users enter sex, age, lifestyle factors (e.g., smoking status), clinical stage, grade, and genotypes of 287 single nucleotide polymorphisms (SNPs), the 5-year survival can be predicted to evaluate whether the colorectal cancer patient is at high risk of death. For example, with the 5-year survival in PLCO cohort as a reference (77.1%), the patient with a predicted survival of 98.5% is considered as low-risk of death (**Figure R4**). Besides, we also emphasized the current limitation as follows: “*Note: This clinical tool is developed preliminarily, with the need of additional validation and improvement in the future. The current results should be interpreted with caution.*”

We have added the above details in the “About” page of the CRC-SPS website.

Figure R4. Example of the predicted survival probability on the “Colorectal cancer survival prediction” page.

5. The construction of PPS287 was valuable for the clinical application, please provide more details (e.g., allele frequency, effect size in different ancestries and heterogeneity statistics) in the Supplementary Tables or CRC-SPS webservice About page.

Response: As shown in the **Table R1**, we have added the detailed information of 287 variants in the **Supplementary Table 4**, which can also be searched on the “Colorectal cancer survival summary statistics” page of the CRC-SPS website.

Minor:

1. It may be interesting to investigate the association between PPS and some population characteristics (e.g., sex, age).

Response: Thank you. We have added the association between some population characteristics and PPS in the ZJCRC and PLCO cohorts of testing stage (**Figure R5**). Although most factors showed no associations with PPS in both two cohorts, the most significant relationships were observed for clinical stage or grade, demonstrating that patients with high stage or grade might be carried with higher PPS (*i.e.*, higher genetic risk of death).

Figure R5. The association between some population characteristics and polygenic prognostic score (PPS) in (A) ZJCRC and (B) PLCO cohorts.

Note: PLCO, Prostate, Lung, Colorectal and Ovarian Cancer Screening Trial.

2. The authors used a previous case-control GWAS (35145 cases and 288934 controls) to evaluate the risk effect of two novel survival-associated loci, please provide more population details in the Method.

Response: We have added more population details about this case-control colorectal cancer GWAS in the Method as follows:

“In addition, we applied a colorectal cancer GWAS meta-analysis of case-control studies to evaluate the risk effect of genome-wide significant prognostic variants. The meta-analysis was performed with totally 35,145 cases and 288,934 controls of EAS and EUR ancestries, derived from NJCRC (1,316 cases and 2,207 controls; EAS), BJCRC (932 cases and 966 controls; EAS), SHCRC (1,116 cases and 1,054 controls; EAS), ZJCRC (1,046 cases and 1,184 controls; EAS), BioBank Japan Project (BBJ; 7,062 cases and 195,745 controls; EAS), GECCO (21,608 cases and 20,278 controls; EUR) and PLCO (2,065 cases and 67,500 controls; EUR) GWASs. More details have been described in our previous study (Ref.1).” in page 16.

Reference:

Ref.1: Risk assessment for colorectal cancer via polygenic risk score and lifestyle exposure: a large-scale association study of East Asian and European populations. Genome Med. 2023 Jan 24;15(1):4.

Reviewer #2 (Remarks to the Author):

This paper reports the development of a polygenic predictive score for bowel cancer survival. Survival GWAS have been performed with two datasets (NJCRC and UKB, Chinese and European ancestry respectively) with subsequent meta-analysis, identifying two loci with suggestive associations, though neither had clear functional associations. PPS were then developed the same cohort, and performance compared in a validation cohort of European individuals (TCGA). The best performing PRS was then tested in cohorts of Chinese (ZJCRC) and European (PLCO) ancestry, and combined with a healthy.

Overall, although a clearly planned methodology and results are presented, there are significant methodological limitations and an overly-optimistic interpretation of the strength of the findings and their impact on clinical practice, which is not supported by the data presented. In addition, important components of the methodology are absent from the paper, as detailed below. Addressing these issues would significantly strengthen the paper.

Response: We thank *Reviewer #2* for the summary and valuable comments. We have revised our study, point by point to address these issues.

In particular, (i) we have added more explanations about the combined model, including its development, validation, clinical application and current limitations, as well as additional evaluation metrics in indicating the significant value of PPS. (ii) We have provided more details in the Method section, and addressed some other questions throughout the manuscript.

1. *I would suggest that that number of cases included does not constitute “Large scale” GWAS meta-analysis. As noted, there is insufficient power to detect GWAS-significant SNP associations. This is an inevitable limit of the available datasets which should be made clearer in the discussion. Is there are a reason that existing prognostic GWAS analysis (e.g. Wills et al noted in the discussion) was not included in the meta-analysis? Was this data unavailable, or overlap between datasets?*

Response: We understand the reviewer’s concern. Here, we have removed the sentences about “large scale” across the manuscript. In our study, we included five available cohorts of colorectal cancer patients with both genotyping and clinical information, from NJCRC (1,082 cases), UK Biobank (2,621 cases), TCGA (470 cases), ZJCRC (543 cases) and PLCO (713 cases) cohorts, of which two largest datasets from the NJCRC and UK Biobank cohorts were used for meta-analysis, while other colorectal cancer survival GWAS datasets (e.g., Wills *et al*’ study of 1,926 patients from COIN and COIN-B clinical trials [Ref.1]) were still not published and not overlapped in our datasets. Nevertheless, as suggested by *Reviewer #1*, we have released the summary statistics of our study in the CRC-SPS web server (<http://njmu.edu.cn:3838/CRC-SPS/>) for researchers’ use.

In addition, we have emphasized the limitation in the Discussion as follows: “First, we only included a total of 3,703 colorectal cancer patients (i.e., NJCRC and UK Biobank cohorts) for the survival-based meta-analysis, with the limitation of

statistical power for detecting genome-wide significant loci; thus, more datasets should be included when available in the future.” in page 12.

Reference:

Ref.1: A genome-wide search for determinants of survival in 1926 patients with advanced colorectal cancer with follow-up in over 22,000 patients. Eur J Cancer. 2021 Dec;159:247-258.

- 2. A significant methodological issue in the construction of the PPS is the use of the same dataset for both GWAS and PPS parameterisation. This overlap will result in overfitting of PPS (Wray NR, Yang J, Hayes BJ, Price AL, Goddard ME, Visscher PM. Pitfalls of predicting complex traits from SNPs. Nat Rev Genet 2013;14:507-15. doi:10.1038/nrg3457) and optimism in performance estimates. This is clearly demonstrated by the drop in performance of the top performing PRS shown in the TCGA cohort in which calibration and AUC are quite poor (Supp figure 3) compared to the NJCRC and UKB cohorts (Supp fig 2).*

Response: We apologize for the misunderstanding. In the first derivation stage, we constructed a colorectal cancer survival GWAS of East Asian (EAS, NJCRC) and European (EUR, UK Biobank) ancestry via meta-analysis, to identify survival-associated SNPs for developing candidate polygenic prognostic scores (PPSs) using multiple approaches. To avoid the potential over-fitting of PPS evaluation, we used the independent TCGA cohort to determine the optimal PPS model (*i.e.*, SNPs and parameters selection), as further validated by external datasets of ZJCRC and PLCO cohorts. The analysis framework is following our and other previous studies (*Refs. 1-2*).

We appreciate the reviewer for pointing out the evaluation results of PPS in the training datasets (NJCRC and UK Biobank cohorts), that have been removed (Supplementary Figure 2 in original manuscript) for avoiding the misunderstanding of readers.

References:

Ref.1: Risk assessment for colorectal cancer via polygenic risk score and lifestyle exposure: a large-scale association study of East Asian and European populations. Genome Med. 2023 Jan 24;15(1):4.

Ref.2: Genome-wide polygenic score to predict chronic kidney disease across ancestries. Nat Med. 2022 Jul;28(7):1412-1420.

- 3. I found the use of both a ‘healthy lifestyle score’ (HLS) and ‘traditional model’ in the PLCA cohort confusing. The HLS is constructed as a categorised risk counting model, and is used in the evaluations of association between pathological features/genetic risk and lifestyle. In the construction of the HLS - it is unclear why were these particular predictors were chosen (based on existing literature?), and how were cut off thresholds defined. Separately, the ‘traditional model’ includes some but not all of these lifestyle predictors and this is used as the comparator and base for the combined model. The authors reference a pre-existing lifestyle*

prediction model for overall survival in bowel cancer patients which has been externally validated (Cheng et al) – is there a reason that this was not used as the traditional model?

Response: Apologize for the confusing and thank you for the valuable comments. In the testing stage, we primarily aimed to evaluate the performance of PPS, and its additional predictive value compared to traditional survival model, including some common prognostic factors, such as smoking and drinking status available in both the ZJCRC and PLCO cohorts.

Subsequently, considering that the PLCO cohort included sufficient lifestyle information, we calculated an integrated healthy lifestyle score that included smoking and drinking status, which allow us to propose a survival evaluation framework for identifying the high-death-risk subgroups, at different levels of pathological stage/grade, genetic risk and healthy lifestyle. We have added more explanations in the manuscript.

As shown in the following **Table R2**, the overall performance of two traditional models (model 1: based on sex, age, *smoking status*, *drinking status*, stage and grade; model 2: based on sex, age, *lifestyle*, stage and grade) were similar. Here, the construction of healthy lifestyle score and definition of cut-off values were based on previous studies (*Refs.1 and 2*), of which included most common lifestyle factors, and we kept variables with low missing rate for analysis; given some missing information (*e.g.*, COX-2 inhibitor use) in the PLCO cohort, we did not select all factors included in Cheng *et al*'s study (*Ref.3*).

References:

Ref.1: Risk assessment for colorectal cancer via polygenic risk score and lifestyle exposure: a large-scale association study of East Asian and European populations. Genome Med. 2023 Jan 24;15(1):4.

Ref.2: Healthy lifestyles, genetic modifiers, and colorectal cancer risk: a prospective cohort study in the UK Biobank. Am J Clin Nutr. 2021 Apr 6;113(4):810-820.

Ref.3: Diet- and Lifestyle-Based Prediction Models to Estimate Cancer Recurrence and Death in Patients With Stage III Colon Cancer (CALGB 89803/Alliance). J Clin Oncol. 2022 Mar 1;40(7):740-751.

Table R2. Summary of the performance of traditional models in the PLCO cohorts.

Cohort	Metric	Traditional model (1) ^a	Traditional model (2) ^a
PLCO	AUC ^b	0.807	0.813
	Sensitivity ^b	0.696	0.678
	Specificity ^b	0.780	0.813
	Harrell's C index	0.786	0.789
	R ² _D (%) ^c	47.23%	47.19%

^a The traditional model (1) included sex, age, smoking status, drinking status, stage and grade. The traditional model (2) included sex, age, lifestyle, stage and grade.

^b AUC, sensitivity and specificity at 5-year survival, of which the optimal sensitivity and specificity were selected based on the Index of Union (IU) method (Comput Math Methods Med. 2017:2017:3762651).

^c Royston and Sauerbrei's R^2_D in Cox proportional hazards models.

Note: PLCO, Prostate, Lung, Colorectal and Ovarian Cancer Screening; PPS, polygenic prognostic score; ROC, receiver operating characteristics; AUC, area under the curve.

4. *Handling of missing data noted in Table 1 and several supplementary tables is not reported – was complete case analysis used, or imputation in some form?*

Response: Thanks. In our cohorts, we applied complete case analysis, which has been added in the Method section.

5. *The methodology for the construction of the combined model was not clearly defined in the paper. Line 174 – “we constructed a combined model by integrating PPS287 with several clinical factors for each cohort” is unclear. I presume this was in a Cox-regression analysis? If so how were the PRS and traditional risk predictors combined?*

Response: We apologize for the unclear description. Indeed, this analysis was conducted with a Cox regression model. Generally, the combined model was constructed as follows: $h(t) = h_0(t)\exp(\beta_{PPS}PPS + \beta_i X_i)$, of which the model included PPS and several traditional factors (X_i , e.g., sex, age, smoking and drinking status). We have added more details in the Method section.

6. *Reported statistics are limited to effect sizes (HR), AUC, and KM curves/log rank tests. Reporting of other key metrics such as variance explained for the PPS would be helpful, in addition to more clinically meaningful metrics such as sensitivity/specificity.*

Response: Thank you for the valuable suggestions. We have added more statistical metrics (e.g., Royston and Sauerbrei's R^2_D , sensitivity and specificity) for the comparison between traditional and combined models (Table R3; Refs. 1 and 2), which could help further explain the additional value of PPS in colorectal cancer survival prediction.

References:

Ref.1: Integrating genome-wide polygenic risk scores and non-genetic risk to predict colorectal cancer diagnosis using UK Biobank data: population based cohort study. *BMJ*. 2022 Nov 9;379:e071707.

Ref.2: Improving reporting standards for polygenic scores in risk prediction studies. *Nature*. 2021 Mar;591(7849):211-219.

Table R3 (Supplementary Table 10). Summary of the performance of traditional and combined prognostic models in the ZJCRC and PLCO cohorts.

Cohort	Metric	Traditional model ^a	Combined model ^a
ZJCRC	AUC ^b	0.644	0.699
	Sensitivity ^b	0.642	0.635
	Specificity ^b	0.601	0.673
	Harrell's C index	0.652	0.715

	R ² _D (%) ^c	17.92%	31.89%
PLCO	AUC ^b	0.807	0.834
	Sensitivity ^b	0.696	0.745
	Specificity ^b	0.780	0.760
	Harrell's C index	0.786	0.818
	R ² _D (%) ^c	47.23%	51.22%

^a The traditional model included sex, age, smoking status and drinking status in the ZJCRC cohort; sex, age, smoking status, drinking status, stage and grade in the PLCO cohort. The combined model included traditional factors and PPS.

^b AUC, sensitivity and specificity at 5-year survival, of which the optimal sensitivity and specificity were selected based on the Index of Union (IU) method (Comput Math Methods Med. 2017:2017:3762651).

^c Royston and Sauerbrei's R²_D in Cox proportional hazards models.

Note: PLCO, Prostate, Lung, Colorectal and Ovarian Cancer Screening; PPS, polygenic prognostic score; ROC, receiver operating characteristics; AUC, area under the curve.

7. *Rather than the results of traditional vs combined model performance in the ZJCRC and PLCO cohort showing that the PRS improves performance, I would argue that these two cohorts demonstrate that overwhelmingly the presence of stage and grade are the strongest predictors of survival (AUC of 0.8 compared to 0.63 – Figure 2B/C and Supp Table 8). In Figure 2 – whilst evaluation of calibration of the PRS in the ZJCRC and PLCO cohorts is helpful, the 'traditional' and 'combined' models in these cohorts ought to be near perfect, as these have been developed in these datasets. Whether the addition of PRS to a model containing stage and grade is of any value needs much more careful consideration than the reporting of the increment in AUC. As above, reporting of variance explained would be helpful, but more meaningfully, clinically relevant metrics such as sensitivity/specificity/PPV, and decision curve analysis, would help elucidate what the incremental gain of PRS would be. See Wand et al <https://doi.org/10.1038/s41586-021-03243-6> for reporting standards for polygenic scores in risk prediction studies.*

Response: We appreciate the reviewer for pointing this issue out. Similar with Question #6, we have added more statistical metrics (e.g., Royston and Sauerbrei's R²_D, sensitivity and specificity; Table R3), as well as the decision curve analysis (DCA; Figure R6) in the model evaluation and comparison. The results of DCA indicated that, the combined model could give more net benefit than PPS or traditional model alone across a range of threshold probabilities.

Figure R6 (Supplementary Figure 5). Decision curve analysis for different colorectal cancer prognostic models in the (A) ZJCRC and (B) PLCO cohorts. The figures show net benefit at 5 years of follow-up.

Note: PLCO, Prostate, Lung, Colorectal and Ovarian Cancer Screening; PPS, polygenic prognostic score.

8. *From the data in Table 3 and Figure 3, although the data show an improvement in OS with healthy lifestyles for some categories this is minimal (e.g. high stg/grade and low genetic risk) and I would argue that the results suggest lifestyle has a more marked impact on risk if stage/grade and genetic risk are low, rather than the discussion of the major effect in higher risk groups (discussion Line 267). I note that the 5 year survival is referenced, whilst the median follow-up for the cohort is 4 years which may limit the strength of this conclusion. Potential confounding from changes in lifestyle over the period of follow-up (for example, prompted by a diagnosis of higher stage/grade cancer) should be considered.*

Response: We appreciate the reviewer for the insightful comments. We have additionally highlighted the protective effect of healthy lifestyle on colorectal cancer overall survival (OS) among the patients with low stage/grade and low genetic risk in the Result and Discussion section as follows:

“Especially, among patients with a low stage/grade and a low genetic risk, the overall survival rate ranged from 65.78% (unfavourable lifestyle) to 92.90% (favourable lifestyle; $P = 0.042$).” in page 9. “Interestingly, adherence to a healthy lifestyle could attenuate the risk of death, especially evident among patients with low stage/grade and low genetic risk ($P < 0.05$).” in page 12.

We agree that the above conclusions may be limited by the follow-up time of current PLCO dataset, that has also been acknowledged in the Discussion section as follows:

“In addition, the model performance and benefit of healthy lifestyle maintenance need to be further validated using a larger longitudinal population with sufficient follow-up time and sample size.” in page 13.

We also appreciate the reviewer for pointing out the issue about the confounding factors during follow-up, which was also indicated by *Reviewer #1*. Given the absence of such factors, we acknowledged the limitation in the Discussion as follows:

“Third, the lifestyle or other confounding factors were derived from the baseline questionnaire in the PLCO cohort, which could not reflect the dynamic changes during the follow-up after colorectal cancer diagnosis; thus, more detailed surveillance is also needed.” in page 13.

9. *The combined model has been made publicly available for use ‘in clinical practice’ and is proposed as an important tool for personalised prediction (Line 278). This model has not been externally validated, having been constructed and tested here only in the PLCO cohort, and I have very serious reservations about it being made available as such as a clinical tool. I would welcome its availability for further research. The appropriateness of the PLCO cohort as the reference dataset for a general CRC population as a reference threshold for high/low risk of death is also not considered.*

Response: We understand the reviewer’s concern. In our study design, due to the potential application of polygenic risk score (PRS) in cancer risk prediction but limited in survival prediction (evidenced by our previous findings [*ref.1*]), we aimed to (i) develop a robust PPS for predicting colorectal cancer OS, (ii) to propose a comprehensive colorectal cancer survival evaluation framework by integrating pathologic characteristics, genetic risk and lifestyle, and (iii) to establish an online clinical tool of personalized survival prediction.

As pointed out by the reviewer, our framework has only been tested in the PLCO dataset with complete pathologic, genetic and lifestyle information (also as a reference panel for selecting threshold), not externally validated by more cohorts. Similar with the concern of *Question #8*, we have acknowledged the limitation and the need of additional validation in the Discussion section as follows: *“In addition, the model performance and benefit of healthy lifestyle maintenance need to be further validated using a larger longitudinal population with sufficient follow-up time and sample size.”* in page 13.

In addition, as also suggested by *Reviewer #1*, in the CRC-SPS web server, we added the explanations of “Colorectal cancer survival prediction” function, and emphasized the current limitation as follows: *“Note: This clinical tool is developed preliminarily, with the need of additional validation and improvement in the future. The current results should be interpreted with caution.”*

References:

Ref.1: Prognostic evaluation of polygenic risk score underlying pan-cancer analysis: evidence from two large-scale cohorts. EBioMedicine. 2023 Mar;89:104454.

Some more minor comments:

1. *Line 72 The sentence “However, survival probability....” Doesn’t quite make sense to me*

Response: We have revised the sentence as follows: *“However, the genetic architecture of colorectal cancer survival outcome has not been widely estimated. Noteworthy, survival probability is another critical indicator, that can reflect the tumour burden and prognosis of disease patients”*.

2. *The paper does not report how cancer cases were identified in the cohorts studied (e.g. which field from UKB was used to ID the cases – first reported instance, or registry data?) End of follow-up/censoring dates should also be defined, as cancer data has been regularly updated for UKB. This could be inserted in the supplementary methods.*

Response: Thank you for the valuable comments, we have added more details for each cohort, such as using the National Cancer Registries data to extract colorectal cancer cases in the UK Biobank cohort, which have been provided in the Supplementary Method section.

3. *Line 131 – C+T, RSF etc should be defined when first mentioned.*

Response: Thank you, we have corrected them.

REVIEWERS' COMMENTS

Reviewer #1 (Remarks to the Author):

All my concerns are addressed.

Reviewer #2 (Remarks to the Author):

Thanks to their authors for the thorough response to both reviewer comments and additions to the paper. The methodology is now more clearly described, and important caveats to the results are now included within the paper and supplementary material. The paper is an interesting addition to the CRC-PRS literature.

Some further comments:

It would be helpful to add confidence intervals to the additional metrics reported in Supplementary table 10.

For clarity within the paper, the information about construction of the healthy lifestyle score given in the response ("Here, the construction of healthy lifestyle score and definition of cut-off values were based on previous studies (Refs.1 and 2), of which included most common lifestyle factors, and we kept variables with low missing rate for analysis") should be added to the supplementary methods (not necessarily verbatim)

Thank you for adding the important "interpret with caution" note on the CRC-SPS web server. However, within the conclusion line 297 of the paper "for use as an important tool for personalized survival prediction" still suggests that this is ready for use in patient care, and I would recommend rewording this with similar caution.

I would encourage the authors to also deposit their polygenic score in the PGS catalogue <https://www.pgscatalog.org/>

Response to the Reviewers' comments:

Reviewer #1 (Remarks to the Author):

All my concerns are addressed.

Response: We thank *Reviewer #1* for the positive comments.

Reviewer #2 (Remarks to the Author):

Thanks to their authors for the thorough response to both reviewer comments and additions to the paper. The methodology is now more clearly described, and important caveats to the results are now included within the paper and supplementary material. The paper is an interesting addition to the CRC-PRS literature.

Response: We thank Reviewer #2 for the positive comments.

Some further comments:

- 1. It would be helpful to add confidence intervals to the additional metrics reported in Supplementary table 10.*

Response: Thank you for the valuable suggestions. We have added the 95% confidence intervals in the **Table R1**.

Table R1 (Supplementary Table 9 in revised version). Summary of the performance of traditional and combined prognostic models in the ZJCRC and PLCO cohorts.

Cohort	Metric	Traditional model ^a	Combined model ^a
ZJCRC	AUC (95% CI) ^b	0.644 (0.588, 0.710)	0.699 (0.649, 0.762)
	Sensitivity (95% CI) ^b	0.642 (0.552, 0.695)	0.635 (0.567, 0.737)
	Specificity (95% CI) ^b	0.601 (0.552, 0.691)	0.673 (0.602, 0.746)
	Harrell's C index (95% CI)	0.652 (0.603, 0.700)	0.715 (0.670, 0.760)
	R ² _D (% , 95% CI) ^c	17.92% (10.06%, 26.48%)	31.89% (23.18%, 40.15%)
PLCO	AUC (95% CI) ^b	0.807 (0.751, 0.868)	0.834 (0.794, 0.886)
	Sensitivity (95% CI) ^b	0.696 (0.623, 0.806)	0.745 (0.654, 0.860)
	Specificity (95% CI) ^b	0.780 (0.667, 0.850)	0.760 (0.657, 0.861)
	Harrell's C index (95% CI)	0.786 (0.745, 0.827)	0.818 (0.785, 0.852)
	R ² _D (% , 95% CI) ^c	47.23% (37.80%, 55.28%)	51.22% (42.94%, 58.21%)

^a The traditional model included sex, age, smoking status and drinking status in the ZJCRC cohort; sex, age, smoking status, drinking status, stage and grade in the PLCO cohort. The combined model included traditional factors and PPS.

^b AUC, sensitivity and specificity at 5-year survival, of which the optimal sensitivity and specificity were selected based on the Index of Union (IU) method (Comput Math Methods Med. 2017;2017:3762651). The confidence intervals (CIs) were derived from bootstrap method with 10,000 iterations.

^c Royston and Sauerbrei's R²_D in Cox proportional hazards models.

Note: PLCO, Prostate, Lung, Colorectal and Ovarian Cancer Screening; PPS, polygenic prognostic score; ROC, receiver operating characteristics; AUC, area under the curve.

- 2. For clarity within the paper, the information about construction of the healthy lifestyle score given in the response ("Here, the construction of healthy lifestyle score and definition of cut-off values were based on previous studies (Refs.1 and 2), of which included most common lifestyle factors, and we kept variables with low missing rate for analysis") should be added to the supplementary methods (not necessarily verbatim).*

Response: Thank you for the suggestion. We have added these details in the **Methods** selection as follows: “The construction of healthy lifestyle score was based on our previous study (*Ref.1*), of which included common lifestyle factors, and we kept lifestyle factors with low missing rate for analysis”.

Reference:

1. *Risk assessment for colorectal cancer via polygenic risk score and lifestyle exposure: a large-scale association study of East Asian and European populations. Genome Med. 2023 Jan 24;15(1):4.*

3. *Thank you for adding the important “interpret with caution” note on the CRC-SPS web server. However, within the conclusion line 297 of the paper "for use as an important tool for personalized survival prediction" still suggests that this is ready for use in patient care, and I would recommend rewording this with similar caution.*

Response: We appreciate the reviewer for pointing it out. We have revised the sentence with “for use as a potential tool in personalized survival prediction”.

4. *I would encourage the authors to also deposit their polygenic score in the PGS catalogue <https://www.pgscatalog.org/>.*

Response: Thank you. We have submitted our polygenic prognostic score to the PGS catalog, with the corresponding PGS number (PGS004586) for publication.